# Laser-induced sublimation extraction for cm-resolution multi-species greenhouse gas analysis on ice cores

Lars Mächler[1], Daniel Baggenstos[1,3], Florian Krauss[1], Jochen Schmitt[1], Bernhard Bereiter[1,2], Remo Walther[1], Christoph Reinhard[1], Béla Tuzson[2], Lukas Emmenegger[2], and Hubertus Fischer[1]

[1]Climate and Environmental Physics and Oeschger Center for Climate Research, University of Bern, 3012 Bern, Switzerland
[2]Laboratory for Air Pollution / Environmental Technology, Empa - Swiss Federal Laboratory for Materials Science and Technology, 8600 Dübendorf, Switzerland
[3]Australian Antarctic Division, Kingston, 7050, Australia

**Correspondence:** daniel.baggenstos@unibe.ch

**Abstract.** Precision, accuracy, and temporal resolution are key to make full use of atmospheric trace gas records in ice cores. These aspects will become especially crucial for ice cores that aim to extend the ice core record to the last $1.5$ Myr, i.e., across the Mid Pleistocene Transition (as currently drilled within the European project Beyond EPICA – Oldest Ice Core (BE-OIC)). The ice from this period is expected to be close to bedrock and, due to glacier flow, extremely thinned with 15,000 years of climate history contained in only one meter of ice. Accordingly, for a century-scale resolution, the sample vertical extent must be reduced to a few cm containing only about $1-2$ mL air STP.

We present a novel combined system for the extraction and the simultaneous measurement of $CO_2$, $CH_4$, and $N_2O$ concentrations, as well as $\delta^{13}CO_2$, which achieves a vertical resolution of $1-2$ cm ($3.5 \times 3.5$ cm cross section) with precisions of $0.4$ ppm, $3$ ppb, $1$ ppb and $0.04$ ‰, respectively, in sublimation tests with standard gas over gas-free ice. This is accomplished by employing a directional and continuous laser induced sublimation followed by analysis of the sample gas by quantum cascade laser absorption spectroscopy (QCLAS). Besides the low sample volume requirements and the vertical resolution capabilities, the described method holds additional advantages over previous methods, including the immunity of the highly specific QCLAS analysis to drilling fluid contamination as well as the non-destructive nature of the spectroscopic gas analysis. The combined extraction and analysis system was extensively tested by sublimating gas-free ice with introduction of a standard gas to determine the accuracy and characterize potential artefacts. Moreover, Antarctic ice samples were measured to confirm the measurement performance, covering the range of variability expected in Pleistocene ice and highlighting the vertical resolution capabilities critical for its application within BE-OIC.

## 1 Introduction

Ice cores are unique natural archives preserving valuable information on atmospheric greenhouse gas concentrations ($CO_2$, $CH_4$, $N_2O$) in typically centennial resolution over hundreds of thousands of years. Reconstructing paleo-atmospheric changes in these gases and their isotopic composition has provided valuable insights into their biogeochemical cycles (Blunier and Brook, 2001; Loulergue et al., 2008; Schilt et al., 2010a; Mitchell et al., 2011; Schmitt et al., 2012; Marcott et al., 2014; Bereiter

et al., 2015; Bauska et al., 2018; Nehrbass-Ahles et al., 2020). However, due to the time consuming and technically challenging nature of ice core analyses and the limited sample availability, the records are still restricted in precision and resolution. This

is particularly pronounced for $\delta^{13}CO_2$, where data is only available for selected time intervals during the last glacial cycle and often lacks the required precision to make full use of this parameter (Schmitt et al., 2012; Schneider et al., 2013; Eggleston et al., 2016; Bauska et al., 2018). These limitations will become even more critical with the advent of a new "Oldest Ice" ice core from East Antarctica (BE-OIC) that aims to extend the ice core greenhouse gas records over the last 1.5 Myr. The deepest part of such a core will be highly thinned by glacier flow leading to 15,000 years of atmospheric history being archived in

only 1 m of ice (Fischer et al., 2013). To make full use of this deep ice core archive, new analytical techniques are required that minimize sample size, increase spatial/temporal resolution, and at the same time maximize precision and accuracy of the analyses to be able to reliably quantify past variabilities. A cm-scale resolution is a reasonable target for such a technique, as the mean age difference of 150 years between neighbouring cm-samples in such old, thinned ice is similar to the width of the gas age distribution of East Antarctic ice cores. Note that other processes counteracting signal preservation might become

significant in the deep ice. For example, the increased vertical concentration gradients in highly thinned ice together with the high in situ temperature of such ice close to bedrock are acting to diminish atmospheric variability by gas diffusion in the ice (Bereiter et al., 2014). To quantify and correct such diffusive smoothing will also set new standards on the required precision of the gas analyses.

In the following, we will briefly discuss the prevalent issues in ice core greenhouse gas analyses that still hinder more precise

and higher resolution records. The most common approach to determine $CH_4$ and $N_2O$ concentrations in ice core samples is melt extraction (either with or without refreezing) followed by quantification of the concentrations with gas chromatography or laser spectroscopy (e.g., Oyabu et al., 2020). Alternatively, $CH_4$ concentrations can be measured in continuous flow analysis (CFA) campaigns with typical depth resolution of $5-10$ cm (Stowasser et al., 2012; Chappellaz et al., 2013; Rhodes et al., 2017). The discrete sample method requires a sample size of $20-40$ g of ice (Schilt et al., 2010a; Mitchell et al., 2011;

Baumgartner et al., 2014; Ryu et al., 2018; Schmidely et al., 2021), corresponding to a typical depth resolution of several cm. $CH_4$ records from melt extraction techniques of discrete ice samples can be affected by production of $CH_4$ during extraction (in extractu) observed in dusty glacial ice from Greenland (Lee et al., 2020; Muehl et al., 2022). CFA records of $CH_4$ (Stowasser et al., 2012; Chappellaz et al., 2013; Rhodes et al., 2017) appear not to be subject to this issue, but suffer from accuracy problems due to the solubility of $CH_4$ in the melt water stream.

For $CO_2$, any contact with liquid water during the extraction procedure must be strictly avoided to circumvent chemical reactions (for example acid/carbonate reactions or oxidation of organic carbon compounds) because they affect the measured $CO_2$ content. For this reason, the analysis of $CO_2$ and $CH_4/N_2O$ were mostly separated in the past despite the larger amount of sample required for both analyses. $CO_2$ measurements are typically performed using mechanical extraction devices (crushing, milling, grating, etc.) combined with either optical gas analysers (e.g., NIR or laser spectroscopy) or gas chromatography (GC)

(Lüthi et al., 2010; Bereiter et al., 2013; Marcott et al., 2014; Nehrbass-Ahles et al., 2020; Shin et al., 2020). The Bern cracker, Grenoble mill or the Bern Centrifugal Ice Microtome (CIM) achieve a vertical sample size of $2-5$ cm (corresponding to as

low as 15 g of ice). Typical $CO_2$ precisions achieved, using these methods, are $1-3$ ppm. The sample throughput ranges from a few up to 20 measurements per day in case of the Bern CIM.

These dry extraction techniques release less than 100 % of the gases trapped in the ice (Bereiter et al., 2015) and have varying extraction efficiencies for air bubbles and different clathrate sizes, which leads to systematic effects in $CO_2$ mainly in and below the brittle ice zone (Lüthi et al., 2010). The only dry extraction technique for $CO_2$ from ice core samples that enables 100 % extraction efficiency is sublimation (Schmitt et al., 2011). While current sublimation extraction methods with a throughput of $1-3$ samples per day are slow in comparison to the CIM, the unfractionated recovery of air enables reliable and precise $\delta^{13}CO_2$ analyses also in partially or fully clathrated ice (Schmitt et al., 2011; Schneider et al., 2013). The interconnected limitation of sample size and precision is especially pronounced for $\delta^{13}CO_2$. Previous work investigating $CO_2$ ice core samples for their carbon isotopic composition using gas chromatography-mass spectrometry (GC-MS) or dual inlet mass spectrometry (MS) achieved reproducibilities of $0.06-0.08$‰ (Schmitt et al., 2011; Schneider et al., 2013) on samples as small as 30 g of ice. Bauska et al. (2014) obtained $0.02$‰, however requiring 10-fold larger samples. All of these methods required separation of $CO_2$ (either via GC or cryogenic trapping) from other gases for the analysis. Finally, interference by drill fluid traces in extracted air samples may affect the gas analyses. Such drill fluid contamination becomes more frequent in the brittle and the very deep ice, where mechanical fracturing enables the contaminant to penetrate the ice core.

To achieve the necessary resolution for the oldest ice while avoiding all issues outlined above, we developed a novel extraction and analysis method using continuous sublimation extraction of ice core sections coupled to a custom-made dual-quantum cascade laser absorption spectrometer (QCLAS, Bereiter et al., 2020). In particular, by using sublimation, the new method avoids the water phase that leads to in extractu formation of $CO_2$ and $CH_4$ and ensures a 100 % extraction efficiency for all gases. Furthermore, the QCLAS is insensitive to drill fluid traces and is designed to simultaneously measure $CO_2$, $CH_4$ and $N_2O$ concentrations as well as $\delta^{13}CO_2$ directly from $1-2$ mL of extracted air (Bereiter et al., 2020). In the following, we present the technical details of the developed method comprising the laser induced sublimation extraction (LISE) system, the vacuum system, the standard gas inlet, and the calibration procedure. While the QCLAS is described in detail by Bereiter et al. (2020), its particular use and the exact measurement procedure, including modification to the inlet and the calibration/-correction procedure for the ice core air sample measurements, are described here. A detailed evaluation of the LISE using this measurement setup follows and finally, we analyzed samples from well characterized ice cores (EDC and Talos Dome) to demonstrate the excellent performance of the LISE/QCLAS system.

## 2 Laser sublimation extraction method

In this section, we describe the laser induced sublimation extraction (LISE) system, starting with the sublimation apparatus, followed by the vacuum system, the standard gases, and then the LISE measurement procedure itself.

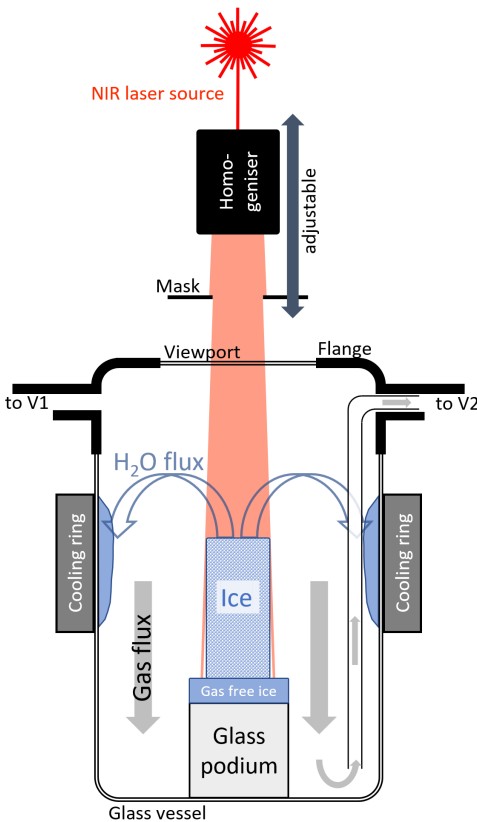

**Figure 1.** Schematic view of a cross section through the extraction vessel during sublimation. The NIR laser beam induces sublimation at the top ice surface. A small fraction of the beam misses the ice sample and hits the underlying gas-free ice block, which prevents further reflection within the vessel. The resulting water vapour is efficiently removed by freezing to the walls of the vessel, while the released gases are evacuated out of the vessel via a glass tube from the bottom of the vessel.

## 2.1 Sublimation extraction

The core of the LISE consists of a 2 L extraction vessel, in which a rectangular block of ice is placed (Fig. 1). The vertical dimension of the ice block is typically 9 cm, which is sufficient for up to 6 consecutive sub-samples. The cross section is variable and can be adapted to the dimensions of the ice samples, but typically a cross-section of $10 - 15\,\mathrm{cm^2}$ is required to achieve a vertical sample resolution of $1 - 2\,\mathrm{cm}$. The air in the ice is released through sublimation of the ice block from top to bottom by vertical illumination with a 150 W near-infrared (NIR) light source. The resulting water vapour is frozen to the cooled walls of the extraction vessel, which leads to low vapour pressure and hence high sublimation efficiency. Consecutively, the released air is cryogenically trapped in dip tubes cooled to 15 K.

The laser (BrightLase®Ultra-500 diode laser, QPC Lasers) emits at a wavelength of 1530 nm. At this wavelength, the penetration depth in ice is less than 1 mm (Warren and Brandt, 2008), which prevents subsurface melting and minimizes

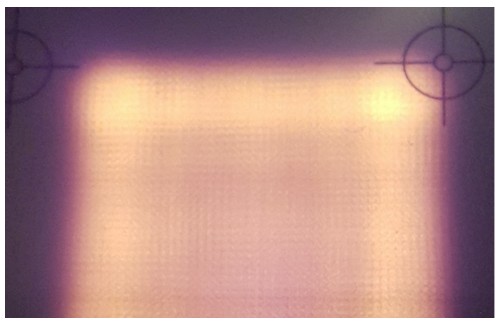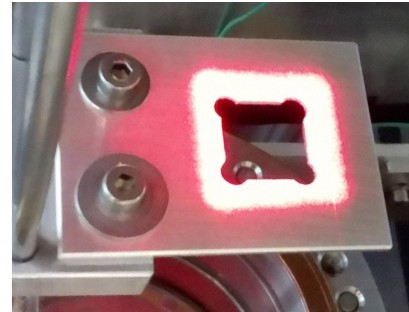

**Figure 2.** Left: Cross section of the laser beam (at 1530 nm) after passing through the homogenisation unit projected on a VRC4 IR detector card showing somewhat higher power densities at the edges. Right: Mask used to further shape the beam. The beam size of the visible trace laser is about 5 % larger than that of the NIR laser.

sublimation of the sides of the ice block (see Sect. 2.3.1). Further, the wavelength is within the transmission regime of common viewports on borosilicate/pyrex basis. The sublimation of a 1 cm high $3.5 \times 3.5\,\mathrm{cm}^2$ ice cuboid requires ca. 33 kJ, which corresponds to 20 min irradiation with 25 W or $2\,\mathrm{Wcm}^{-2}$. The 150 W maximum optical power of the laser therefore gives a large margin for losses of light before absorption by the ice. The laser beam is transmitted via a fiber optic patch cable (MHP550L02, Thorlabs) and has initially a Gaussian spatial power distribution. An optical homogenisation unit (Bayerisches Laserzentrum GmbH) is used to widen and homogenize the beam, and to reshape it into a quadratic cross section, while a specifically shaped aperture further forms it to match the ice block cross section (Fig. 1 & 2). The unit consists of a collimator, two arrays of cylindrical lenses that split the incoming beam into different beamlets, and a Fourier lens that superposes the beamlets at its focal length of 25 cm into a $4 \times 4\,\mathrm{cm}^2$ cross section (e.g., Zimmermann et al. (2007)). The optical system is mounted on a XY linear translation stage, the position of which can be adjusted in the vertical direction. A visible (635 nm) trace laser facilitates the alignment.

The extraction vessel (Fig. 1) consists of a sealed glass vessel with a flat base and lid (approx. 100 mm diameter) with a kovar glass adaptor to a ConFlat flange (DN100 CF). A CF spacer flange with two DN16 CF ports connects the vessel to the rest of the vacuum line. The top flange, where the laser beam enters the vessel, is a DN100 CF viewport (Pfeiffer), which can be removed for loading ice samples.

In principle, ice sublimates at any pressure below the triple point of water (6 hPa). However, we empirically found that the pressure in the extraction vessel should be kept below 0.2 hPa to avoid the build-up of needle-like sublimation structures (further on referred to as "Penitentes", see Sect. 2.3.1). Cooling the walls of the glass vessel with a temperature controlled (ca. $-120\,^{\circ}$C) aluminium ring removes most of the vapour by freezing it out onto the vessel wall. The ring is cooled by supplying liquid nitrogen (LN2) drops to its inner compartment which is filled with copper swarfs (size ca $2 - 3$ mm, see also Schmitt et al. 2011). The swarfs increase the surface area and tortuosity, thus facilitating evaporation of the LN2 rather than pooling at the bottom of the ring. Taking into account the different thermal contraction coefficients of aluminium and glass, we left a gap of about 1 mm between ring and glass at room temperature. The ring and the glass part of the vessel are placed in a dewar,

and the gap between the dewar edge and the vessel is covered by a thermal insulator (polyurethane foam). A PID controller (diaTRON 116, Jumo) regulates the LN2 supply and stabilizes the temperature of the ring within $0.2\,°C$.

All the metal surfaces of the extraction vessel were coated with gold to reduce adsorption and desorption of gases, prevent oxidation, and in particular to suppress any $CO_2$ formation from potential carbon impurities in the steel walls and welds. Despite the gold plating, desorption of gases still occurs when the walls get warmer during the sublimation by scattered NIR light. Accordingly, the top flanges are cooled to $-15\,°C$ by an air stream at $-110\,°C$ of pressurised air cooled via a copper heat exchanger mounted in a $2\,L$ dewar (Schmitt et al., 2011) to minimize such effects. In a similar fashion to the cooling ring, LN2 is dripped into the dewar, at a rate regulated by a PID controller (West 2300, UK) to reach and maintain the targeted temperature (measured by a thermocouple) of the exiting air stream.

The entire extraction system, consisting of homogenizer, extraction vessel, and LN2 supply, is surrounded by an aluminium casing for laser safety. Part of the casing can be removed for loading of the sample, refilling of LN2 or readjusting of the homogenizer position.

## 2.2 Vacuum system and extraction line

Once the gases are released from the ice, they are pushed (similar to a diffusion pump) downwards by the vapour stream that originates from the ice surface to the cooled outer surfaces of the vessel where the water vapour freezes (Fig. 1). To prevent gas accumulation/fractionation at the bottom of the vessel downstream of this "water vapour pump", we evacuate the vessel through an L-shaped glass tube from the bottom of the extraction vessel. The upper end of this tube is connected to one of the exit ports of the middle flange leading to the extraction line that consists up to V3 of 3/8 inch stainless steel tubing (bold line in Fig. 3). The gases follow the pressure gradient caused by freezing the residual water vapour flux in a second, external water trap at $-100\,°C$, and finally the dry gas is collected in dip tubes cooled to $15\,K$ in a helium exchange cryostat acting as a cryopump at the end of the line.

The external water trap (Fig. 3) consists of a $10\,cm$ long cooled aluminium body assembled around 3/8 inch tubing. Again, the temperature of the aluminium body is regulated within $0.2\,°C$ using a PID controller (diaTRON 116, Jumo) via a LN2 supply to bores filled with steel wool in the aluminium body. The cryostat (CCS-500/204, Janis) used to cool the dip tubes to $15\,K$ is an exchange gas closed-cycle cold trap system with six dip tube ports. Each dip tube consists of a sealed off 1/4 inch SilcoNert® coated tube connected with a VCR connector to a full metal valve (Fujikin, FUDDFM-71G-6.35). Six dip tubes are attached to a custom made aluminum manifold (hereafter called Hexatubus, photo in Appendix FigureA1) that connects with a 1/4 inch tube to the rest of the extraction line at V3 for consecutive automated filling. The six dip tubes are consecutively filled with the gas released from the ice sample. After the extraction the entire Hexatubus is transferred to the QCLAS system (see Sect. 3) for the gas analysis.

Besides removing the released gas from the extraction vessel, the middle flange of the extraction vessel also allows to introduce standard gases at a rate similar to the gas release rate of the ice core samples, to mimic the extraction process and quantify potential blank contributions or fractionation processes. To this end the second port of the extraction vessel connects

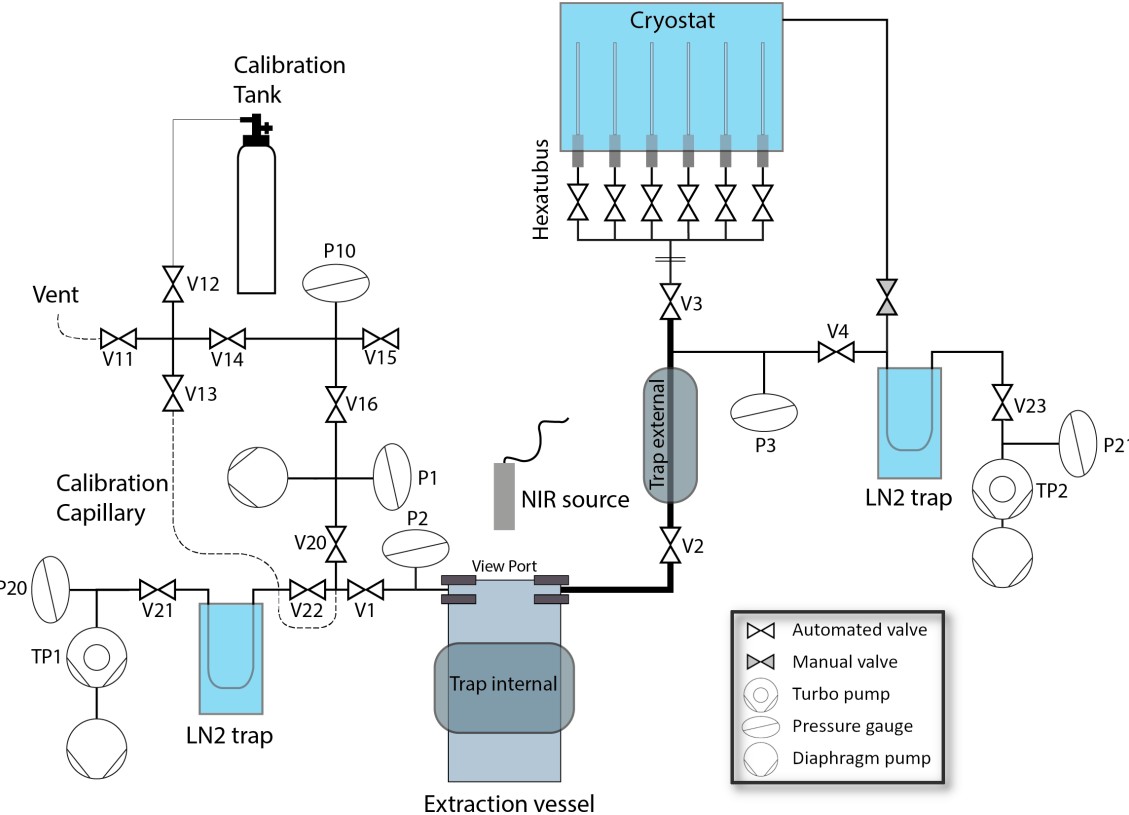

**Figure 3.** Schematic diagram of the laser induced sublimation extraction (LISE) system. Tubing size upstream of V21 and V23 is mostly 1/4 inch, except in the sample pathway indicated by bold lines (3/8 inch) and upstream of V12 (1/8 inch). Individual components are described in the text.

via valve V1 to the outlet of the standard capillary that is used to introduce standard gases (see Sect. 2.4) to the extraction system (Fig. 3).

All valves in contact with sample or standard gas at low pressure, i.e. V20,V22,V1,V2,V3, and V4, are full metal pneumatic valves (Fujikin FWBR-71 series). All seals in the extraction system are either CF copper or VCR stainless steel gaskets. The high vacuum is maintained by two pump stations, each equipped with a turbomolecular pump (Turbo HiPace 80, Pfeiffer) backed by a diaphragm pump (MVP 015-4, Pfeiffer). A similar diaphragm pump is used to evacuate the standard gas inlet and in a first stage the extraction vessel to a pressure below 1 hPa, which is measured using a Pirani gauge (P1, TPR 280, Pfeiffer). Both pump stations are connected to the extraction line via a LN2 trap that removes any remnant water vapour, hence improves the vacuum, and assures that the pressure monitored at the turbomolecular pump reflects essentially dry gas pressure. The pressures P20 & P21 at the pump stations are measured by a Pirani/cold cathode combination gauge (PKR 361, Pfeiffer). In the extraction line, we installed capacitance gauges (P2, CMR 363, & P3, CMR 364, Pfeiffer) due to their suitable materials exposed to the vacuum side, i.e ceramic and stainless steel, and good accuracy with low zero drift.

## 2.3 Characterization of the LISE

### 2.3.1 Sublimation front

The net sublimation rate can be described as a modified Hertz-Knudsen equation, and hence it is proportional to the difference between the vapour pressure above the ice surface and the equilibrium vapour pressure at ice temperature, with a proportionality coefficient that strongly decreases towards lower temperature (Kossacki et al., 1999). With a heat source (here supplied by the NIR laser) the ice temperature will rise and increase the sublimation rate until a steady state is reached at the surface or it starts to melt. Hence, to sustain sublimation the vapour needs to be transported away and frozen at the internal trap. For a top surface sublimation speed of $0.05 - 0.1\,\mathrm{cm\,min^{-1}}$, the surface of the ice needs to have a temperature of $-27$ to $-20\,^{\circ}\mathrm{C}$, thus, far below the melting point. The heat input from the laser beam necessary to sustain such sublimation rates is $2 - 4\,\mathrm{W\,cm^{-2}}$ and almost independent of ice temperature as sublimation heat only varies within $1\,\%$ over a sublimation temperature range from $0$ to $-100\,^{\circ}\mathrm{C}$ (Feistel and Wagner, 2007).

The cooling process of the ice during the pumpdown phase before the NIR laser is switched on is mainly caused by the heat loss due to sublimation and decreases the temperature of the ice surface until the loss reaches a steady state with the thermal radiation from the surrounding vessel. The residual flow of air liberated from the ice is about two orders of magnitude smaller than during the extraction, depending on both the size of the ice sample (total surface vs. irradiated surface) and of the laser power. Thus, during the pumpdown process, we expect about a $5\,\%$ loss of the ice volume, which also efficiently removes any adsorbed gases from the ice sample surface.

Irradiating the top surface of the ice increases its temperature and hence the sublimation rate, but also induces a temperature gradient across the ice. A temperature gradient of $10\,\mathrm{K\,cm^{-1}}$ in the ice block removes around $2 - 3\,\mathrm{W}$ from the sublimation front as the thermal conductivity of ice ranges from $2.2\,\mathrm{W\,mK^{-1}}$ at $0\,^{\circ}\mathrm{C}$ to $3.2\,\mathrm{W\,mK^{-1}}$ at $-100\,^{\circ}\mathrm{C}$ (Fukusako, 1990). While such a heat loss can be easily compensated by higher irradiation power, it may lead to increased temperature on the flanks of the ice inducing unwanted sublimation at these locations, thereby curving the ice surface and thus decreasing the effective resolution of the extraction.

To attain a uniform and horizontal sublimation front, we aimed at a homogeneous square beam shape. Despite the homogenizer beam shaping, however, the power distribution still exhibited higher density at the edges of the square (roughly $50\,\%$ higher intensity in the outer 2.5 mm, Fig. 2) and led us to use an aperture as a mask to cut off the outer energy maxima leading to a more homogeneous illumination of the top surface. The mask with side length of 1.5 cm was placed 9 cm below the Fourier lens. However, we empirically found that using a square mask the ice at the corners of the ice block retreated more slowly than the main surface, resulting in $1 - 2\,\mathrm{cm}$ high ice peaks at each corner after 6 cm of sublimation. We adjusted the aperture design to increase irradiation at the corners of the ice by drilling 3 mm holes centered at each corner of the mask, which improved the flatness of the sublimation front considerably.

### 2.3.2 Stacked ice resolution experiments

The vertical resolution of the gas extraction that can be achieved with this setup was investigated by preparing ice consisting of stacked sections alternating between gas-free ice (FI) and bubble ice (BI). Observing the released gas flow while sublimating over the ice sections reveals the blurring of transitions and whether side sublimation of lower sections contribute to the gas flow. We prepared a 9 cm long ice sample consisting of stacked $1 \times 3.5 \times 3.5 \, cm^3$ sections alternating between FI and BI, and another 7 cm long ice sample consisting of alternating 2 cm sections of the same lateral dimensions. The bubble ice sections were cut from the Byrd ice core (1797 m depth, tube Nr 1147 C) with an air content somewhere between 0.10 to $0.11 \, mL_{STP}g^{-1}$ (Raynaud and Whillans, 1982). The sections were "glued" together by carefully distributing a small amount of Milli-Q water on the surface before stacking them. Even though Milli-Q water is not gas free, its contribution to the measured air flow is minimal.

The ice was then sublimated as described in Sect. 2.5 with a laser power of 100 W down to ca. $1/4$ of the bottom-most bubble ice section. The released gases increase the pressure above TP2 detected by P21 (Fig. 3), which is proportional to the gas flow as all water vapour has been removed by the traps. A rough calibration of the P21 pressure/flow relationship was achieved by admitting gas through the standard gas capillary at known flow rates into the system. The transitions from FI to BI are used as depth markers and enable conversion from time to length. Note that the conversion is assumed linear, which leads to an overestimation of distance in the last ice section due to lower laser intensity.

The extracted air flow vs. sublimated depth of the 1 cm stacked ice (Fig. 4) follows a slightly deformed periodic square wave with resolved peaks, demonstrating that the achieved resolution is better than 1 cm. The minima between peaks, i.e., during sublimation of FI sections, do not decrease with depth, indicating that only gases released due to side sublimation of the following BI section significantly contribute to the gas flow, but not of the lower BI sections. The gas flow during the FI section is, therefore, the sum of the decreasing flow from the previous and the increasing flow from the following BI section. While in the 1 cm-stack the gas flow from two BI sections do mix in the middle part of a transition through a FI section, in the 2 cm-stacked ice both contributions reach the background level before the start of the sublimation of the next BI, indicating that the two peaks are fully separated (Fig. 4). The steeper slope of flow from BI to FI than from FI to BI is a result of gas emission due to side sublimation. The emission is not present in the former as the (FI) ice sublimated on the side is gas free. In both transitions, the non-uniform sublimation fronts contribute to the blurring of the transition. The non-uniform sublimation front results roughly into an exponential increase or decrease when sublimated over a transition with a half-value depth of $0.1 - 0.2$ cm. The side sublimation contribution increases almost linearly within 0.5 cm to around 20 % of peak height approximately $0.2 - 0.3$ cm before peak height, where the exponential increase from gases released at the sublimation front starts to become significant. Note that the pump time required to reach 25 % of initial pressure is around 100 s, corresponding to 0.1 cm of sublimated ice; mixing after release does, therefore, not significantly decrease resolution. In conclusion, two single pulses can be distinguished within $0.2 - 0.4$ cm (roughly double the distance in which the signal drops to 50 % of its initial value) and are fully separated at a spatial distance of 1 cm. While we have shown that the contribution of side sublimation from

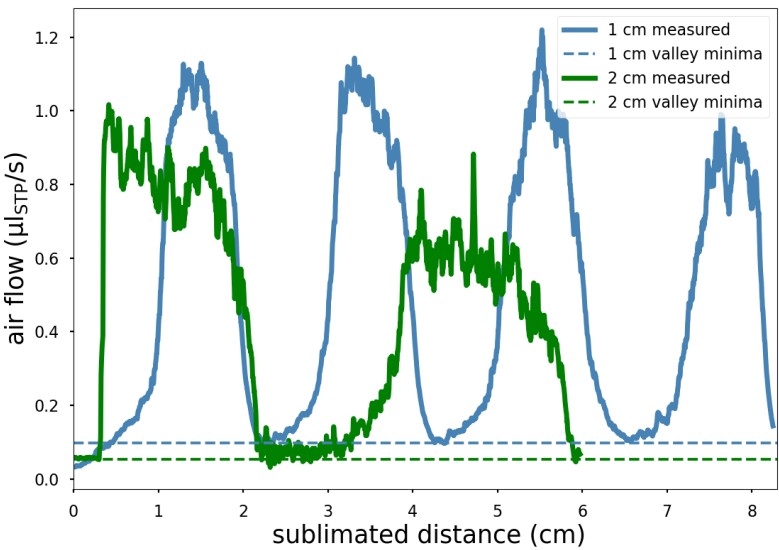

**Figure 4.** Released air flow (solid lines, "measured") during sublimation (100 W laser power) of alternating stacked gas free ice (FI) and bubble ice (BI). Blue is alternating 1 cm ice layers, starting with FI. Green is 2 cm layers starting with BI. The dashed lines mark comparable minima during the sublimation of FI that form extraction flow "valleys". For the sublimated distance a conversion factor of 1 cm per 1050 s is used.

lower layers is small (a few % at most), it may still lead to some smoothing across the sub-samples if the concentration gradient
in a sample is large.

### 2.3.3   Penitentes

A faster sublimation is beneficial considering the potential contamination from gases desorbed from the surfaces. While optical power loss can be compensated by using a higher laser power, a phenomenon appearing on the sublimating surface that might impact resolution is the occurrence of penitentes structures. In nature, penitentes are needle-like sublimation ice structures at
much larger scale found in high-altitude dry mountain regions, e.g. in the Andes (Bergeron et al., 2006). The formation of such jagged ice spikes is linked to a surface instability driven by the combination of sublimation of the ice and reflection of the sun light by surrounding features. The redistribution of incident radiation into initially small depressions in the ice leads to locally higher sublimation rates and further increases the surface depressions. This positive feedback mechanism can result in nature in penitentes up to several meters in height with tilts toward the predominant direction of solar radiation (Bergeron et al., 2006;
Cathles et al., 2014).

We observed similar pentitentes-like structures of much smaller size appearing and evolving in time that negatively impact the sublimation front and thus limit the vertical resolution of the extraction. Their appearance was observed only when the

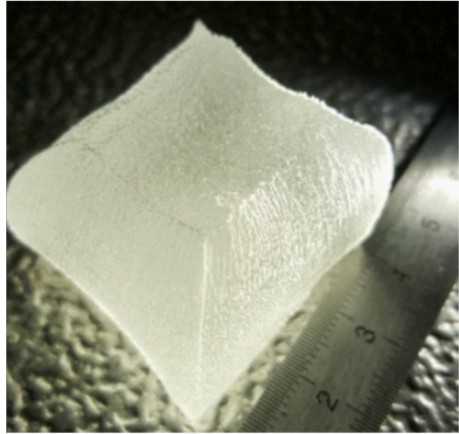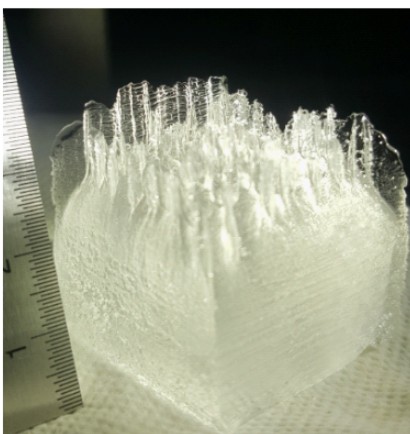

**Figure 5.** Ice after $4\,\mathrm{cm}$ of sublimation. Penitentes in the right picture occur at pressures of more than $0.15 - 0.2\,\mathrm{hPa}$ measured at the top of the extraction vessel, while on the left the pressure was kept below $0.15\,\mathrm{hPa}$.

pressure measured at gauge P2 was higher than $0.2\,\mathrm{hPa}$ (Fig. 5). The size of such penitentes can exceed $1\,\mathrm{cm}$ after $3 - 4\,\mathrm{cm}$ of sublimation and "grow" further. The pressure threshold slightly depends on the gas mixture in the vessel, as in gas-free ice the
245 transition to the build-up of penitentes is lower than for bubble ice. We speculate that a higher pressure inhibits efficient water vapour transport away from the ice surface to play a major role in the build-up of penitentes (Berisford et al., 2021). Setting the internal trap temperature to $-120\,^{\circ}\mathrm{C}$ and restricting the sublimation speed to below $1.0\,\mathrm{cm}$ per $20\,\mathrm{min}$, keeps the pressure in the extraction vessel below $0.15\,\mathrm{hPa}$. With this configuration the formation of pentitentes can be avoided.

### 2.4 Standard gas inlet and calibration gases

The principle of identical treatment (Werner and Brand, 2001) of sample and reference material cannot be completely adhered to in our case, due to the lack of a suitable ice reference material for gases. Thus, similar to Schmitt et al. (2011), we approximate the identical treatment by admitting a calibration gas into the extraction vessel while sublimating gas-free ice. The flow rate is adjusted to match the gas flow released from ice core samples. Subsequent steps are kept identical to the ice core sample procedure (for details, see Sect. 2.5). The standard gases from pressurized cylinders are first introduced to the
standard venting chamber via V12 up to a pressure set by the respective regulator at the cylinder (depending on the selected flow rate around $0.1\,\mathrm{MPa}$ over-pressure). A vent capillary (fused silica, $100\,\mu\mathrm{m}$ i.d., $0.5\,\mathrm{m}$) keeps the system flushed at a rate of $1 - 2\,\mathrm{mL_{STP}\,min^{-1}}$ to avoid isotopic fractionation. The capillary (fused silica, $50\,\mu\mathrm{m}$ i.d., $1\,\mathrm{m}$), through which the gases are admitted into the extraction line, induces a laminar flow. Due to the four order of magnitude smaller pressure at the outlet, this flow is only dependent on the pressure P10 at the high-pressure end.
For the calibration of the LISE (see Sect. 4.1) and the QCLAS (see Sect. 3.2), and to link the $CO_2$ isotope ratios derived by QCLAS to the international Vienna PeeDee Belemnite (VPDB) scale, we prepared a set of calibration gases (Table 1) with specific mixtures covering the full range from the preindustrial to glacial atmospheric composition in $CO_2$, $CH_4$, and $N_2O$ and

**Table 1.** Reference values of custom-made standard gases prepared for the calibration of the QCLAS from Bereiter et al. (2020), # 1-7; and reference air obtained from INSTAAR, # B4 & B3. The concentrations and isotope ratio values cover the range of expected variations found in air samples from ice cores. The standard #2 is matrix air (free of $CO_2$ and $CH_4$) that was used to generate the various standard mixtures by adding different amounts of trace gases. The uncertainty of the values are given in parentheses as 1-sigma.

| Cylinder # | $CO_2$ (ppm) | $CH_4$ (ppb) | $N_2O$ (ppb) | $\delta^{13}CO_2$ (‰) |
|---|---|---|---|---|
| 1. | 248.8 (0.01) | 528.6 (0.05) | 242.0 (0.250) | −6.56 (0.037) |
| 2. | 0.2 (0.01) | 0.1 (0.04) | 172.7 (0.100) | n.a. |
| 3. | 157.7 (0.02) | 331.2 (0.04) | 189.2 (0.310) | −3.64 (0.015) |
| 4. | 167.0 (0.02) | 811.0 (0.03) | 340.4 (0.020) | −9.80 (0.017) |
| 5. | 345.5 (0.03) | 779.8 (0.14) | 325.7 (0.040) | −3.62 (0.023) |
| 6. | 239.2 (0.03) | 527.2 (0.08) | 236.4 (0.280) | −6.58 (0.020) |
| 7. | 341.8 (0.05) | 339.2 (0.07) | 167.4 (0.330) | −10.00 (0.014) |
| B4 (CB10176) | 279.51 (0.05) | - | - | −8.73 (0.03) |
| B3 (CB10855) | 191.74 (0.06) | - | - | −8.67 (0.07) |

a range of $-3.54$‰ to $-10$‰ in $\delta^{13}CO_2$. The production procedure and further details are discussed in Bereiter et al. (2020). Note that Standard #1 and #6 have similar isotopic composition within $0.02$‰ and $CO_2$ concentrations, which are close to the values expected in ice core samples and are therefore used as the preferred working standards for characterization of the LISE and as QCLAS reference gases. Two additional air reference cylinders with preindustrial atmospheric composition in $CO_2$, however with present-day $\delta^{13}CO_2$ signature (Table 1), were used as independent target to determine the long-term accuracy and precision of the QCLAS. These cylinders were prepared at the Stable Isotope Lab at the Institute for Arctic and Alpine Research (INSTAAR, University of Colorado, USA).

## 2.5 LISE procedure

To avoid contamination of the ice core sample during processing, the extraction vessel (EV) is rinsed after each use with ethanol followed by ultrapure water and baked out at $195\,°C$ overnight in a vacuum oven. The EV is then remounted to the extraction apparatus. Prior to an extraction, the EV is evacuated again overnight combined with heating to $100\,°C$, to remove water vapour and potential contamination from the inner surface and passivate the metal surfaces once more. The six dip tubes of the Hexatubus are placed into the (warm) cryo-cooler and attached to the system. The dip tubes and the extraction line are evacuated with a turbomolecular pump overnight reaching pressures below $10^{-7}$ hPa.

The extraction procedure starts by turning on the He cryostat, as it takes more than two hours to reach the 15 K needed to trap the gas components in the dip tube. The EV is isolated from the extraction line and filled with argon gas (Ar, ALPHAGAZ 1) to avoid ambient air entering the system during the ice sample loading. The internal water vapour trap around the EV is

cooled down to $-25\,°C$ before loading the EV with the ice core sample. During the cool-down time the ice core sample is prepared.

The ice core sample is cut from an ice core section with a band saw in a cold room at $-20\,°C$. To minimize the risk of contamination, a layer of at least 5 mm is removed from each surface of the bulk ice core sample. Furthermore, the surfaces of the ice core sample in contact with the band saw are further shaved with a stainless steel microtome knife. After this procedure, the weight of a 9 cm tall ice sample amounts to $70-80\,g$. To load the ice core sample, the viewport is unmounted while a steady flow of Ar limits the invasion of laboratory air into the EV. First, a pre-cooled glass cube ($6 \times 5 \times 5\,cm^3$) is placed into the EV, followed by a layer of gas-free ice ($2 \times 5 \times 5\,cm^3$) that prevents the NIR laser beam from hitting the glass cube. The ice core sample is then added and aligned to the NIR laser beam with the help of the visible trace laser. For all handling steps polyethylene gloves are worn over regular gloves to reduce potential contamination. After re-mounting the viewport and sealing the EV, the Ar-flow is switched off, and a diaphragm pump is used to evacuate the extraction volume via V1 and V20. Meanwhile, the internal water vapour trap is set to $-70\,°C$ and the metal flange parts are cooled to $-15\,°C$ with an external cooling ring. After 30 min of pumping, the (predominantly water vapour) pressure in the extraction vessel reaches 0.1 hPa and a turbomolecular pump takes over the vessel evacuation. The internal trap temperature is then lowered to $-120\,°C$, while the external water trap, in the line connecting the EV to the Hexatubus, is set to $-100\,°C$ hindering any water vapour emanating from the EV from reaching the dip tubes. After another 30 min of turbomolecular pumpdown, the pressure at P20 reaches $2 \times 10^{-6}$ hPa. During the entire pumpdown procedure, $1-2\,mm$ of ice on each surface is sublimated away, which acts as an additional cleaning step (see also Sect. 2.3.1).

After finishing the pumpdown, V1 is closed, and the gas stream is routed into the extraction line via V2. The sublimation is initiated by setting the NIR laser to 100 W. Note that due to radiation losses only about 30 % of the power is available to sublimate ice (see Sect. 2.3.2). To stabilize the gas flow and equilibrate the extraction line surfaces, the liberated sample gas is wasted into the turbomolecular pump for the first 10 min of the sublimation process. A longer pumping time did not improve the results and leads to unnecessary loss of sample. Afterwards, collection of the gas sample starts by consecutive opening and closing of the six dip tubes for 25 min each (corresponding to ~1 ml STP each). By integrating the readouts of the pressure gauge P3 we can estimate the total amount of gas collected in one dip tube with an accuracy of 1%. With six collected air samples from a 9 cm ice core sample the vertical resolution is 1.5 cm. After completing the filling procedure, the entire Hexatubus is detached from the extraction system and removed from the cryo-cooler. The trapped sample gases inside the dip tubes are left for at least one hour to homogenize prior to the QCLAS measurement.

The standard gas over gas-free ice procedure only differs from the above in that during the sublimation process standard gas is introduced through the capillary into the extraction vessel at a rate similar to the one released from an actual ice sample. All pressure readings and valve switches are performed automatically by a custom-made LabVIEW program. The only task that needs to be performed manually during the sublimation is to periodically lower the homogenizer unit of the NIR laser to keep the distance to the top of the sublimating ice sample comparable.

## 3 QCLAS gas analysis

The QCLAS system, which performs the actual measurements of greenhouse gas concentrations and the isotopic composition of $CO_2$, is described and discussed in detail by Bereiter et al. (2020). In this section we provide a short summary of its main characteristics, the measurement procedure, and describe additional modifications that were necessary to ensure the fidelity of the (isotopic) analysis of small air samples derived from ice core sublimation. A series of measurements of reference gases over 6 months were used to assess the spectrometer's long-term stability and reproducibility.

### 3.1 Upgraded QCLAS system

The key elements of the system are the dual-laser concept for simultaneous measurement of $CO_2$, $CH_4$, and $N_2O$ concentrations as well as $\delta^{13}CO_2$, the custom-developed, low-volume multipass absorption cell (MPC), custom-made high-performance data acquisition and laser driving electronics, as well as the vacuum manifold that allows a non-fractionating admission of standard gases or samples into the absorption cell. A multi-port selector (6-port, Vici Valco Instruments Inc.) with individual vents allows switching between the standard gas cylinders while maintaining an uninterrupted gas flow. A calibration scheme was implemented to derive transfer functions that relate the instrument response to externally calibrated standard gases for a range of pressures in the absorption cell. For the calibration the standard gases #3−#7 in Table 1 are used, which span the full range of expected values in $CO_2$, $CH_4$, and $N_2O$ concentrations as well as $\delta^{13}CO_2$ isotopic composition in ice cores.

With this setup, we demonstrated sample-standard gas repeatabilities ($1\,\sigma$) for discrete gas samples of $1\,mL$ STP amount (corresponding to $10-15\,g$ ice) of $0.03\,ppm$ for $CO_2$, $2.2\,ppb$ for $CH_4$, $1\,ppb$ for $N_2O$ and $0.04\,\text{‰}$ for $\delta^{13}CO_2$ (Bereiter et al., 2020). Further we showed that the laser spectroscopic approach is immune against potential contaminating compounds found in drilling fluids.

To carry out the standard gas admission and mixing described below, the gas handling system (see Fig. 6) has been modified from Bereiter et al. (2020) and now includes two mass-flow controllers (analyt MTC, 358 Series) along with a critical orifice ($20\,\mu m$ hole diameter) and a capillary flow restriction, respectively. Fast switching between gases with the multi-port selector has proven inadequate since it takes more than $10\,min$ for the trace gas concentrations to stabilize after a switch. Thus, a direct line circumventing the multi-port selector has been added, allowing the admission of two different standard gas streams into the MPC in rapid succession. This second line enables the analysis of different standard gases within minutes of each other, as required by the measurement procedure, and also improves the determination of the calibration functions by providing an unchanging reference, whilst the calibration gases are cycled through via the multi-port selector.

### 3.2 QCLAS measurement procedure

To achieve the highest accuracy for the dip tube analysis we use a pointwise offset correction approach. This is required to account for spectrometer related artifacts causing dependencies of the retrieved $\delta^{13}CO_2$ values on gas pressure and $CO_2$ amount fraction. We decided to minimize these effects rather than just correct for it, mainly because we observed a variability of the dependencies over time. The dip tube measurement is preceded and followed by standard gases in the following manner

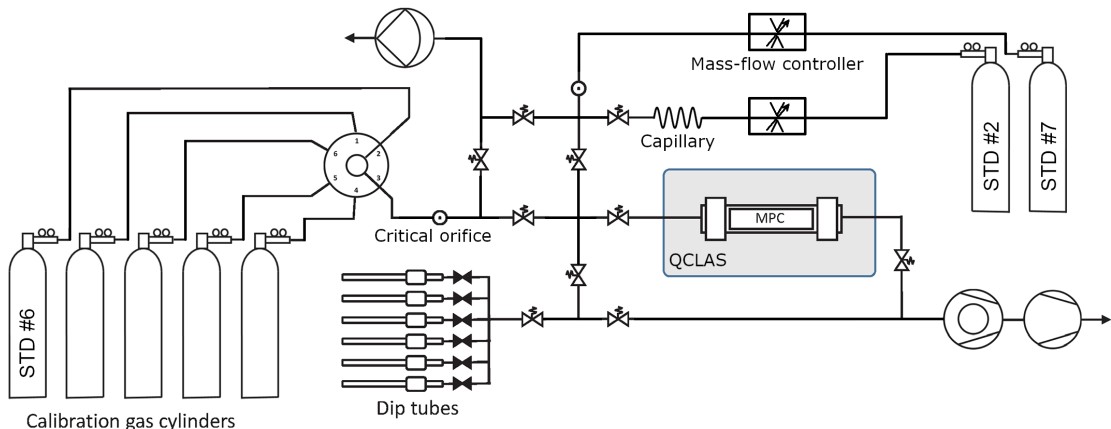

**Figure 6.** Gas handling system used to introduce the small air samples from dip tubes into the QCLAS. The entire system is made of UHV stainless-steel tubing and uses exclusively VCR seals in the low pressure part downstream of the flow restrictions (critical orifices and fused silica capillary) up to the multi-pass cell. The valves are pneumatically actuated, full metal valves (Fujikin Inc., Japan).

(Fig. 7): First an aliquot of standard gas #6 is admitted into the absorption cell at $5.5\,\mathrm{hPa}$, which serves as a conditioning step and control measurement. Then the MPC is fully evacuated and the sample gas is expanded from the dip tube into the MPC. After another full evacuation the MPC is filled with a standard gas mixture (#2 & #7) that is concentration-matched within 1 ppm and pressure-matched within 1 Pa to the preceding sample. This standard is used later to pointwise correct the concentration-specific offset in $\delta^{13}CO_2$. The concentration matching is achieved by mixing a constant gas flow (approximately $5\,\mathrm{mL\,min^{-1}}$) of standard #7 with a variable gas flow of the $CO_2$-free standard #2 (Fig. 6). The parameters of the mass-flow controller for standard #2 are calibrated beforehand and cover a range from 150 to 300 ppm. Finally, a pressure-matched aliquot of standard #6 is admitted into the cell. This standard is used later to pointwise correct any offsets in the concentration values. The entire cycle of standard and sample sequences for each dip tube takes less than 20 min.

Each measurement block lasts for $80\,\mathrm{s}$, with the signal being averaged over the entire period. These raw data are calibrated using previously determined calibration curves (according to Bereiter et al. (2020)) and are denoted as $[X]_{cal}$. The next step is to apply a drift correction to the sample measurement using the value of the immediately following standard measurements. The drift corrected values of the sample concentrations $[X]_{SA,cal,pdc}$ (with $SA$ for sample and $pdc$ for pointwise drift correction), using the calibrated concentrations of the standard #6 pressure matched measurements ($ST6$), are calculated in the following way:

$$[X]_{SA,cal,pdc} = [X]_{SA,cal} + ([X]_{ST6,ref} - [X]_{ST6,cal}) \tag{1}$$

where $[X]$ stands for the $CO_2$, $CH_4$, and $N_2O$ concentrations, and $ST6,ref$ refers to the externally established reference concentrations of standard #6 from Table 1.

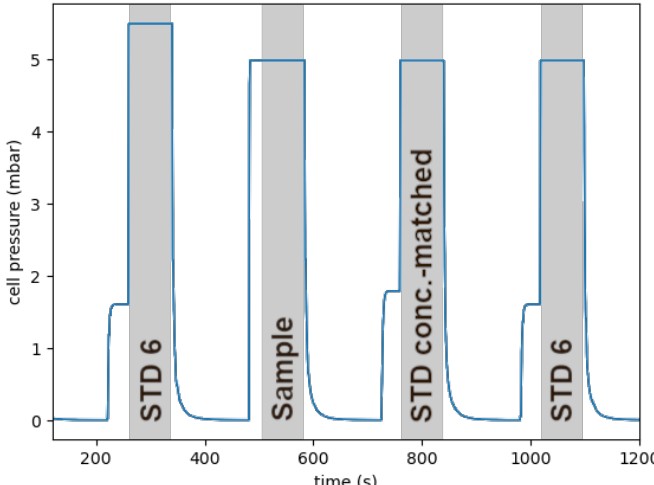

**Figure 7.** Measurement procedure of a single dip tube sample as recorded in the pressure time-series inside the absorption cell. The sample is flanked by several standard gases in a bracketing approach. The shaded blocks indicate measurement periods. The 'shoulder' of approx. 1.7 hPa before every standard gas (STD) period indicates a 30 s purge to equilibrate all surfaces before the cell is filled to a specified pressure.

For $\delta^{13}CO_2$, the correction takes a similar form:

$$\delta^{13}C_{SA,cal,pdc} = \delta^{13}C_{SA,cal} + (\delta^{13}C_{STcm,ref} - \delta^{13}C_{STcm,cal}) \tag{2}$$

where *STcm* stands for the concentration-matched standard gas mixture (the third block in Figure 7).

### 3.3    Long-term stability and precision

The long-term stability of the QCLAS system is verified by repeatedly measuring two standard gases (B3 and B4 in Table 1) over the period from September 2021 to March 2022 in approximately two-week intervals for a total of 49 individual measurements of each gas. These standards are well suited to mimic ice core derived ancient air samples, as they are not used in the

calibration, making them truly independent targets, and their $CO_2$ concentrations span the range expected in actual samples (see Sect. 2.4). The results are displayed in Fig. 8. For $CO_2$, the drift correction improves the repeatability by about factor two, i.e. from 0.2 ppm to better than 0.1 ppm. In absolute terms, the stated externally calibrated values of B3 and B4 are 0.1 to 0.2 ppm higher than our measurements, which is acceptable for the intended ice core analyses.

For $CH_4$, the drift correction has no clear effect on the repeatability. The long-term repeatability is about 3.5 ppb, but could

be better if not for a few outliers. These outliers can be identified, however, using the secondary beam path in the QCLAS that is directed through a reference cell and monitored with an additional IR detector. (Bereiter et al., 2020). As B3 and B4 were not independently referenced for $CH_4$ or $N_2O$, the accuracy of our measurements cannot be assessed with them.

For N$_2$O, the drift correction improves the precision for B3 and B4 to 1 to 1.5 ppb, but is again impacted by outliers. Especially for B4, there is a conspicuous bias in the first half of the measurement series whose cause we haven't been able to identify.

The largest impact of the pointwise drift correction is on the $\delta^{13}$CO$_2$ values. The repeatability improves from 0.49‰ to 0.03‰ for the concentration-matched standard. For the two standards B3 and B4 we found within our precision an identical value in $\delta^{13}$CO$_2$ of $-8.713$‰, which is in good agreement with the externally determined isotopic composition from Table 1.

Overall, the repeatability of the measurements, while slightly worse for concentrations than those of Bereiter et al. (2020), is sufficient for the intended ice core analyses. All reported measurements in Sect. 4 include the drift correction.

## 4 Ice measurements

### 4.1 Standard over ice experiments

In this section we examine sublimation experiments where calibration gases (see Table 1) are continuously admitted into the extraction vessel while sublimating a piece of gas-free ice (Fig. 9). This mimics the extraction with real ice samples closely and is supposed to reveal anomalies in the studied gas concentrations that arise from the process of sublimation such as: i) production by reflected or stray laser radiation hitting the metal flange, ii) co-deposition of trace gases with the freezing water vapour on the side of the extraction vessel, iii) fractionation in the water vapour flux from the sublimating ice sample, iv) ad- or desorption at the glass and metal walls in the presence of water vapour, and v) potentially other unknown effects.

We found that for all target parameters the biases to the expected values are small, with average anomalies of $+0.9$ ppm for CO$_2$, $-0.11$‰ for $\delta^{13}$CO$_2$, $+2.9$ ppb for CH$_4$, and $+0.3$ ppb for N$_2$O. Furthermore, the anomalies are not significantly different for the three standard gases, which span the range of concentrations expected from real ice samples and for $\delta^{13}$CO$_2$ even exceed that. The magnitude of these biases suggests that the aforementioned processes alter the composition of the sample gas only marginally. The standard deviation calculated over all data shown in Fig. 9 amounts to 0.42 ppm, 3.1 ppb, 1.1 ppb and 0.043‰, respectively for CO$_2$, CH$_4$, N$_2$O and $\delta^{13}$CO$_2$. These represent a conservative estimate of the precision for a single ice core-derived measurement.

Considering the CO$_2$ measurements, for which a very high precision is achieved, a consistent and reproducible pattern is observed: an elevated concentration in the first dip tube followed by a lower concentration value in the second dip tube. For $\delta^{13}$CO$_2$, a trend towards more depleted values with increasing sample number was found. It is not clear yet what causes this behaviour, but apparently it is inherent to our system. A contamination by insufficient pumping has been ruled out. In Fig. 11 we show results for the Talos Dome ice samples with and without a finger-specific bias correction, which clearly show an improvement of the homogeneity of the results for CO$_2$ and $\delta^{13}$CO$_2$.

When accounting for individual dip tube average biases, the standard deviations of the standard over ice measurements improve to 0.29 ppm and 0.031‰ for CO$_2$ and $\delta^{13}$CO$_2$. As we are using averages of 7 (CO$_2$, CH$_4$, N$_2$O) and 5 ($\delta^{13}$CO$_2$) individual experiments, the biases are reasonably confined, but their quantification can be further improved in the future.

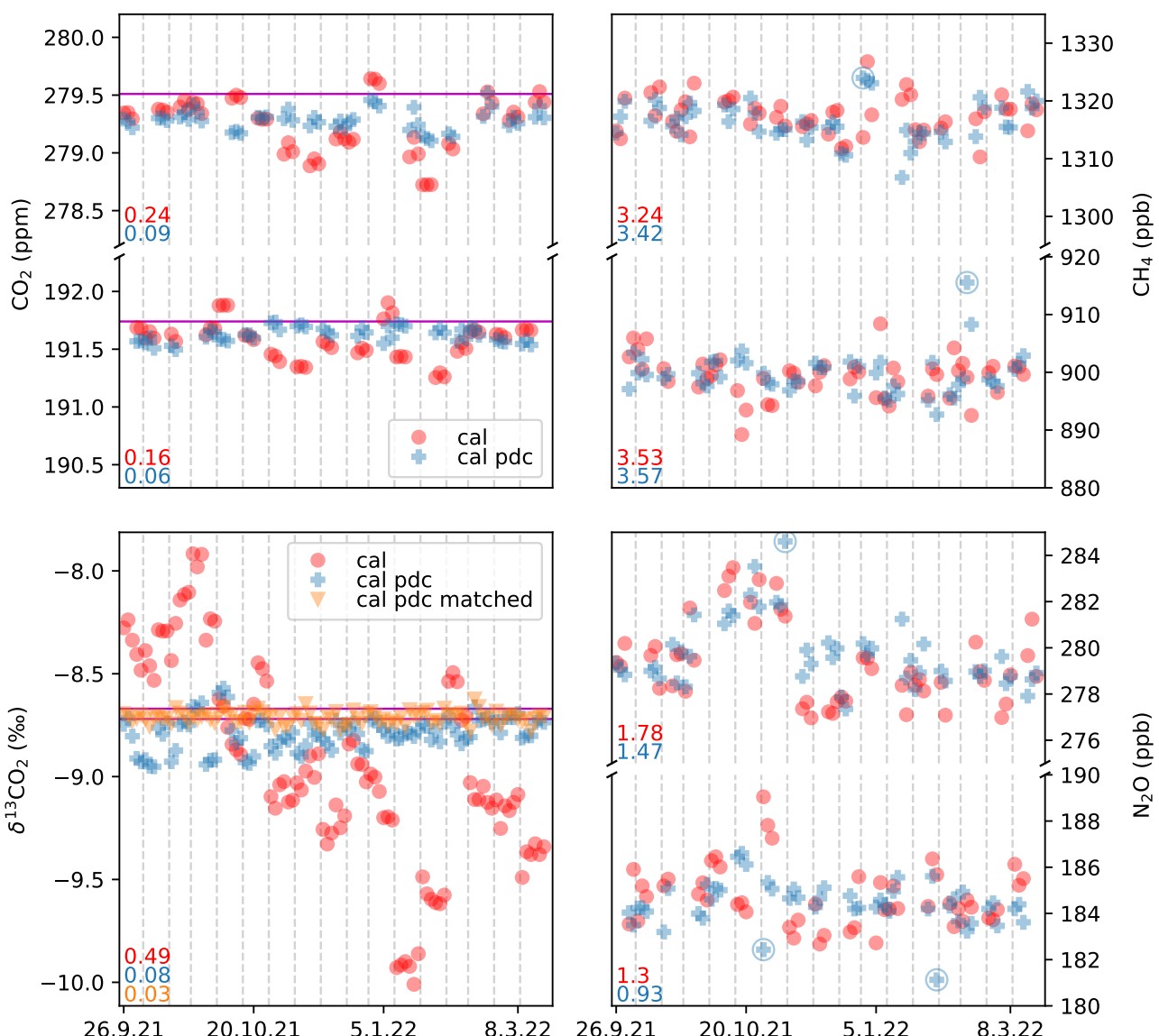

**Figure 8.** Repeated bi-weekly measurements (n = 49) of B3 and B4 over 6 months. The two standards differ significantly in all measured concentrations, but are essentially identical in $\delta^{13}CO_2$ (Table 1). Each measurement set, separated by the dashed gray lines, is composed of 3 aliquots of each standard. Data are shown as calibrated only (cal) as well as calibrated and drift corrected (cal, pdc) according to Eq. (1) and (2). For $\delta^{13}CO_2$, drift-corrected values using the concentration-matched standard (cal pdc matched) are shown additionally. On the lower left of each subplot, the standard deviation (1-sigma) achieved with the different corrections are shown. For $\delta^{13}CO_2$, the standard deviation is shown for the combined measurements of both standards. The circled symbol denotes an outlier identified in the secondary ('reference') channel (more details in the main text). Outliers are not removed from the standard deviation, and hence our precision estimates based on these measurements are conservative. The horizontal magenta lines depict the externally determined values as listed in Table 1.

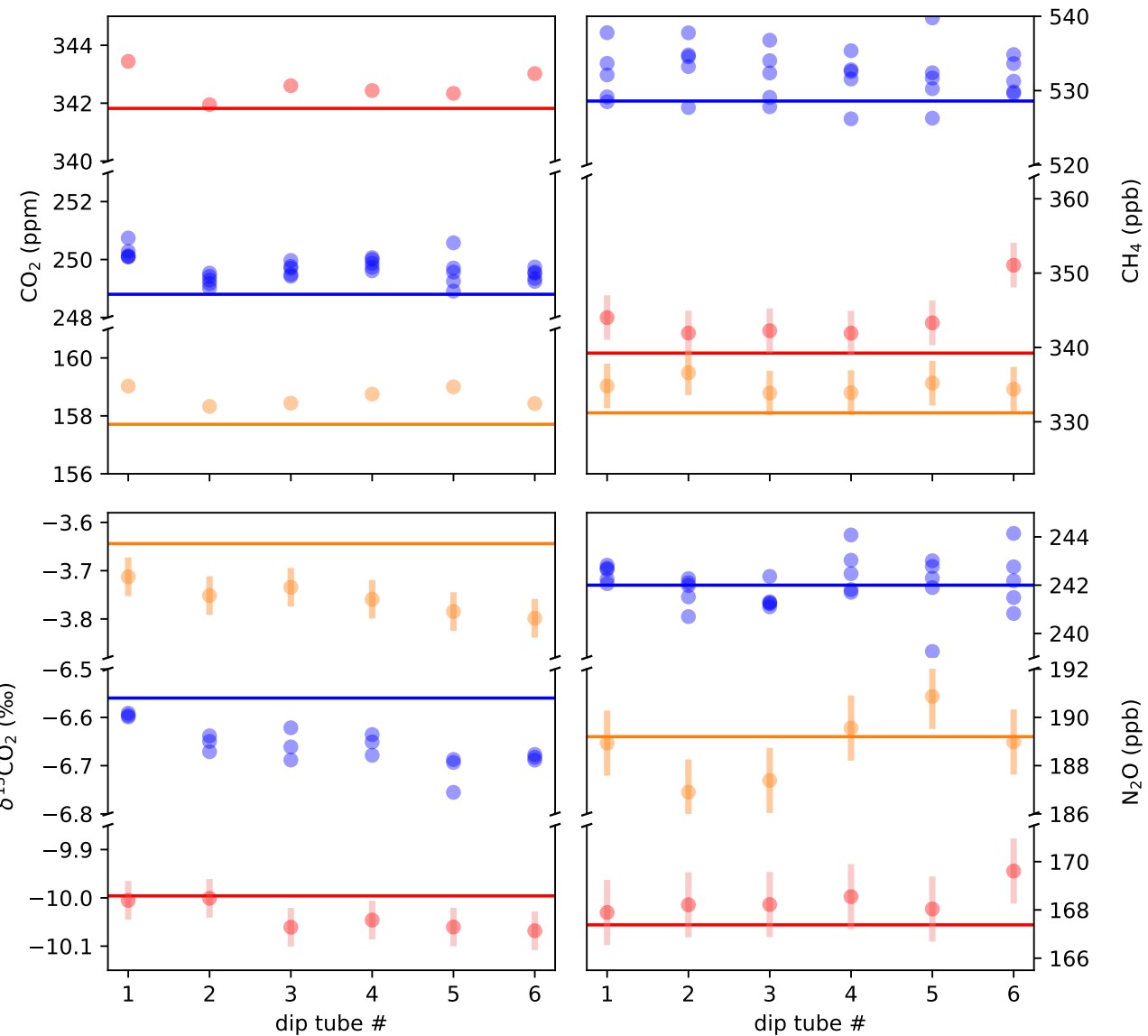

**Figure 9.** The standard over ice experiments characterize the small offsets that are introduced through the extraction process and gas transfer from the extraction vessel to the MPC. The standards #3 (orange) and #7 (red) were measured once each, while #1 (blue) was repeated five times. Error bars show the long-term reproducibility of discrete standards measured by the QCLAS only as determined in Fig. 8. For standard #1 the error bars are omitted to improve legibility. For $CO_2$, the error bars are smaller than the marker size.

## 4.2 Ice core samples

As the final aim of our development is to measure ice core samples, we used individual EPICA Dome C (EDC) samples of glacial and interglacial ages covering the full range of concentrations expected in Pleistocene ice, and one continuous 57 cm long ice section from the Talos Dome ice core to assess the resolution capabilities of the LISE extraction method.

### 4.2.1 EDC

A total of seven EDC ice core samples were measured with the LISE/QCLAS system, three from a glacial and interglacial period, and one additional fully clathrated sample from 1518 m depth (see Table 2 and Fig. 10). For $CO_2$, the measured values agree very well with previous data obtained with different extraction techniques. The variability of $1 - 2$ ppm within a 10 cm long ice sample is noticeably higher than in our standard over ice experiments, reflecting true variability in the concentration in the ice as to be expected from the stochastic nature of bubble trapping at the firn-ice transition. The sample at 768.35 m depth is an outlier in this regard with a much higher intra-sample variability, likely due to the fact that this sample lies in the bubble-to-clathrate transition zone (Neff, 2014), where layered early clathratization occurs and differently fast permeation rates between bubbly layers and clathrate layers for different gas species lead to the cm scale variability in the gas composition (Lüthi et al., 2010; Oyabu et al., 2021).

Similarly, also for $CH_4$ the absolute values agree well with the published record. The intra-sample standard deviation is with $4 - 16$ ppb also exceeding the analytical uncertainty determined above. This variability can likely be attributed to differential layer trapping, which produces noise on the cm-scale (Fourteau et al., 2017). Interestingly, the $CH_4$ concentrations do not seem affected by fractionation in the transition from bubbles to clathrates represented by the 768.35 m sample. This agrees with the high-resolution $CH_4$ measurements by Fourteau et al. (2020), where sections from the transition zone likewise do not show inflated variability.

$N_2O$ is known to be affected by (variable) in situ production during periods of high dust content (Schilt et al., 2010a), which confounds the comparison to other $N_2O$ data during glacial maxima. Note that the Talos Dome ice core is least affected by this process due to its lower dust content (Schilt et al., 2010b; Fischer et al., 2019). Generally, our EDC samples agree well with the published record, but there is a tendency to a positive bias of ∼7 ppb even for interglacial samples not affected by in situ production. A similarly high bias compared to previous literature values has been observed recently by Schmidely et al. (2021) using a refined, more efficient melt extraction technique and may point to non-quantitative extraction of highly soluble $N_2O$ from the meltwater in older studies. The 100 % extraction efficiency of our sublimation method avoids any such solubility issues. The cm-scale variability also observed in interglacial samples is elevated compared to the established analytical performance as for $CO_2$ and $CH_4$, and likely for the same reasons.

The $\delta^{13}CO_2$ has the worst signal-to-noise ratio of our measured parameters, complicating the comparison to existing records. In addition, $\delta^{13}CO_2$ needs to be corrected for gravitational fractionation in the firn column (Craig et al., 1988; Sowers et al., 1989; Schwander et al., 1993), to be able to compare reconstructed atmospheric values from different ice cores. For our samples, the correction is made using interpolated $\delta^{15}N_2$ data from Haeberli et al. (2021). After this correction, the atmospheric

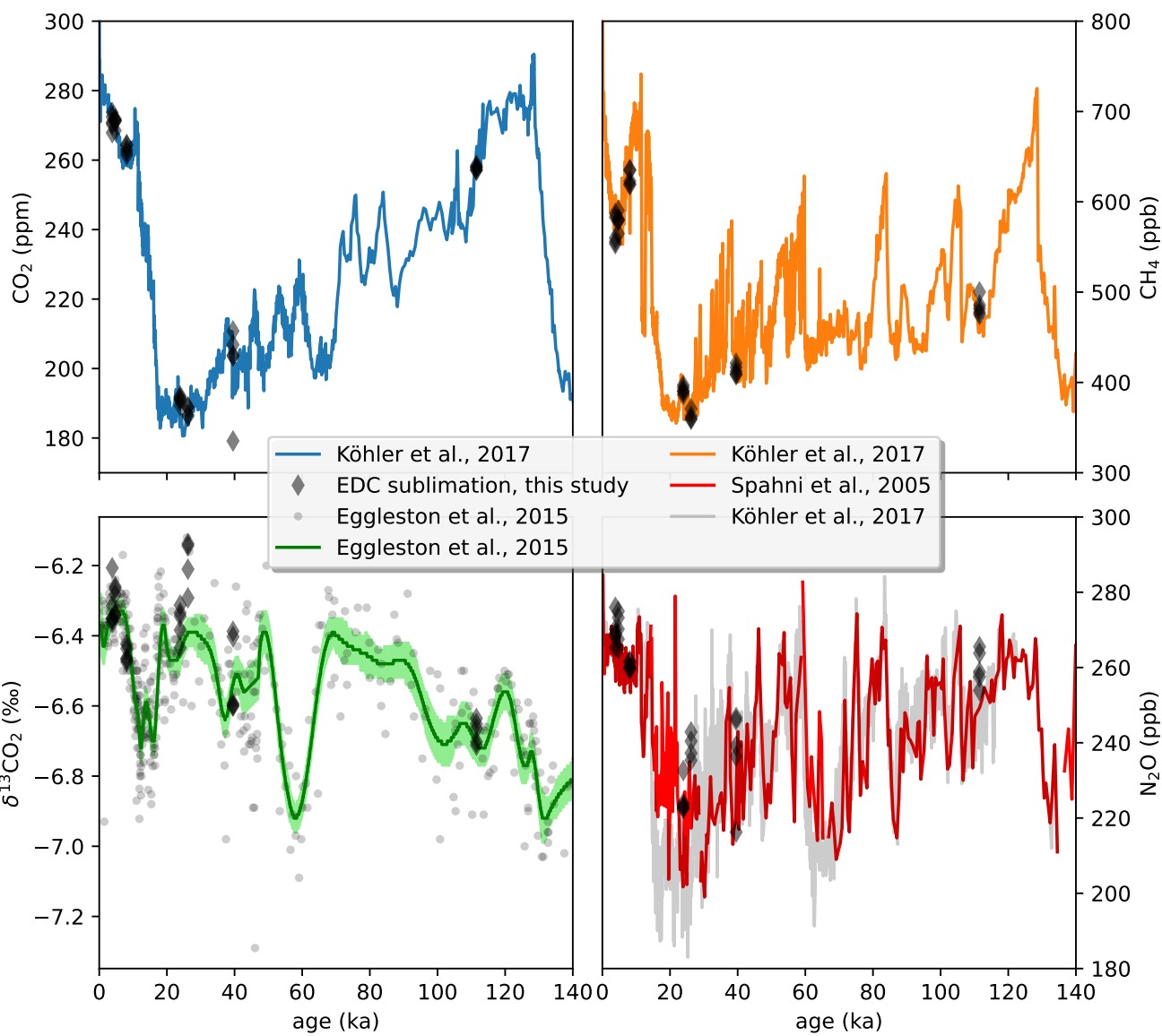

**Figure 10.** Comparison of selected EDC ice core samples (Table 2) with published records. Reconstructed atmospheric $CO_2$ and $CH_4$ data compilations from Köhler et al. (2017). Individual $\delta^{13}CO_2$ data points and a smoothing spline are from Eggleston et al. (2016). $N_2O$ data from Spahni et al. (2005). All LISE/QCLS data are corrected for the extraction biases established by the standard over ice experiments. $\delta^{13}CO_2$ is additionally corrected for gravitational fractionation using interpolated $\delta^{15}N_2$ data from Haeberli et al. (2021).

$\delta^{13}CO_2$ values agree well with previous records. The variability is with 0.02 to 0.07‰ only slightly elevated with respect to the analytical precision except for the sample from the bubble-clathrate transition zone (768.35 m depth, 0.11‰ SD).

**Table 2.** Analyzed EDC samples. Each sample was $\sim 9$ cm long, yielding 4-5 sub-samples. Standard deviations (SD) are 1-sigma and given in parenthesis.

| depth (m) | AICC2012 age (ka) | # of sub-samples | $CO_2$ (ppm) | $CH_4$ (ppb) | $N_2O$ (ppb) | $\delta^{13}CO_2$ (‰) |
|---|---|---|---|---|---|---|
| 203.50 | 3.84 | 5 | 271.1 (2.2) | 567.9 (15.8) | 269.8 (3.9) | −6.32 (0.06) |
| 227.20 | 4.72 | 5 | 270.9 (1.4) | 579.0 (9.2) | 270.2 (4.0) | −6.30 (0.04) |
| 316.80 | 8.12 | 5 | 263.1 (1.3) | 626.4 (8.1) | 260.5 (0.9) | −6.45 (0.02) |
| 598.95 | 23.92 | 5 | 191.0 (1.0) | 391.6 (4.0) | 225.0 (4.3) | −6.36 (0.05) |
| 618.75 | 26.22 | 4 | 187.7 (1.4) | 363.2 (5.3) | 239.0 (3.3) | −6.20 (0.07) |
| 768.35 | 39.54 | 5 | 200.9 (12.5) | 413.7 (5.4) | 236.9 (12.3) | −6.52 (0.11) |
| 1518.00 | 111.50 | 5 | 257.7 (0.6) | 484.1 (9.9) | 259.9 (4.7) | −6.67 (0.03) |

#### 4.2.2 Talos Dome

A second set of validation measurements was performed using ice core from Talos Dome (TALDICE, Urbini et al. (2006)). A 57 cm long section from 1548 m depth, less than 100 m above bedrock, was analyzed continuously leading to 27 individual data points over the entire section. Although initially undated, the deep TALDICE ice core below 1438 m has recently been re-examined using $\delta^{18}O_{atm}$, $\delta D$, and $^{81}Kr$ combined with an inverse ice chronology model to produce the TALDICE-deep1 chronology (Crotti et al., 2021). This close to bedrock, the stratigraphic layers are highly thinned. The thinning leads to the climatic signal getting compressed, and atmospheric variability expected to occur on the decimeter scale. The deep TALDICE ice is thus similar in terms of thinning to the BE-OI core currently being drilled at Little Dome C, where a highly compressed stratigraphy similarly requires high-resolution analysis techniques to fully recover the paleo-atmospheric signal (Fischer et al., 2013). Therefore, these TALDICE ice samples provide a benchmark-test for the new LISE extraction, though the stratigraphy expected for the deep BE-OIC will be even more compressed.

The results (Fig. 11) confirm that the studied section contains carbon dioxide jump (CDJ) 10a, where $CO_2$ concentrations increase from 210 ppm to 219 ppm in less than 250 years, as previously documented in the EDC ice core (Nehrbass-Ahles et al., 2020). In our study, however, this short jump is likely resolved by 15 individual data points (see discussion on age scale below), showing a relatively constant rise rate over this interval except for one slightly elevated sample after the first quarter of the jump. For both $CH_4$ and $N_2O$ there is variability on short timescales that was previously not resolved with very rapid swings in $CH_4$ concentrations of 50 ppb over 5 cm, similar in size to the $CH_4$ changes observed for some of the smaller Dansgaard-Oeschger events during Marine Isotope Stage 3 (Baumgartner et al., 2014; Rhodes et al., 2015). The events in $CH_4$ are also observed in $N_2O$, although more gradual in nature, as expected from the more than ten times longer atmospheric lifetime of $N_2O$ and with an earlier onset of the $N_2O$ increase, as previously observed for Dansgaard-Oeschger events (Schilt et al., 2013). Signs of in situ production such as high uncorrelated scatter of the measurements is not observed during this 57 cm sequence.

Regarding possible corrections for the dip-tube specific biases documented in the standard over ice experiments (Sect. 4.1), there is strong evidence in the TALDICE measurements that indeed both $CO_2$ and $\delta^{13}CO_2$ are affected by the same patterns that were established in Sect. 4.1. For $CO_2$, the first measurement of every subset (separated by vertical lines in Fig. 11) is elevated compared to the following measurements. After subtracting the established standard-over-ice-biases this feature disappears. Similarly for $\delta^{13}CO_2$, there is a decreasing trend in each subset in the uncorrected data, which is also the case in the standard over ice experiments. After applying the standard over ice correction these artifactual trends are removed, leaving a gradual enrichment over the entire 57 cm section. Accounting for a linear trend over the entire 57 cm sequence, the remaining variability amounts to $0.05\,‰$ (1-$\sigma$), just slightly more than the expected statistical noise due to the measurement precision.

Comparison with previous studies is complicated due to the large discrepancy in vertical resolution and thus time (Fig. 12). Only $CO_2$ has a reference record from EDC (Nehrbass-Ahles et al., 2020) that has enough resolution to allow for a direct comparison. For $CH_4$ and $N_2O$ the absolute concentrations broadly agree, but due to the additional variability resolved in our new data as well as the uncertainties in the age scale (see below), it is difficult to make any further interpretation. It does appear that, just like for EDC ice, our $N_2O$ concentrations are again slightly elevated compared to the published data. $\delta^{13}CO_2$ has not previously been measured in this time interval at all.

We point out that on the TALDICE-deep1 age scale (Crotti et al., 2021), our $CO_2$ concentration data does not fit the published record very well. However, there are several caveats to consider: i) TALDICE-deep1 ends at 1548 m depth, which means the age of our data is unconstrained towards older ages, ii) the stated uncertainty of TALDICE-deep1 in this age interval is almost 2000 years, and iii) the tie points at this age are rather poor (their Fig. 3b). All of these lead us to speculate that the gas age-depth curve of TALDICE-deep1 is too steep and slightly too old at this depth compared to EDC on AICC2012 (Veres et al., 2013). To match our $CO_2$ data to the EDC record we need to flatten the TALDICE age-depth curve (by manually adjusting tiepoints), which also aligns better with the $^{81}Kr$ dated horizons (their Fig. 6b).

## 5  Conclusions

We described a coupled extraction and analysis system capable of simultaneous, high-precision measurements of $CO_2$, $CH_4$, and $N_2O$ concentrations, as well as $\delta^{13}CO_2$, on ice core samples. Compared to existing techniques, our LISE/QCLAS system has the following advantages: i) No pre-processing of the sample (separation of individual trace gases) is necessary as the QCLAS measures absorption spectra directly in whole air; ii) The measurement of the carbon isotopic composition of $CO_2$ and the greenhouse gas concentrations takes place on the same sample of extracted air without isobaric interferences common to mass spectrometry, eliminating the need for splitting the sample; iii) Sublimation achieves $100\,\%$ extraction efficiency, which is especially crucial for deeper, semi-clathrated and clathrated ice; iv) Controlled sublimation from top to bottom of a sample reaches a vertical resolution ($1-2$ cm) needed to resolve atmospheric variability in highly thinned ice; v) Equal treatment of ice samples and standard gas over gas-free ice allows us to characterize the entire system and correct for alterations to the studied gases due to the extraction procedure; vi) There is the possibility to reuse the sample gas due to the non-destructive nature of the QCLAS measurement. This is especially helpful for the gravitational correction of $\delta^{13}CO_2$, as $\delta^{15}N_2$ could be measured

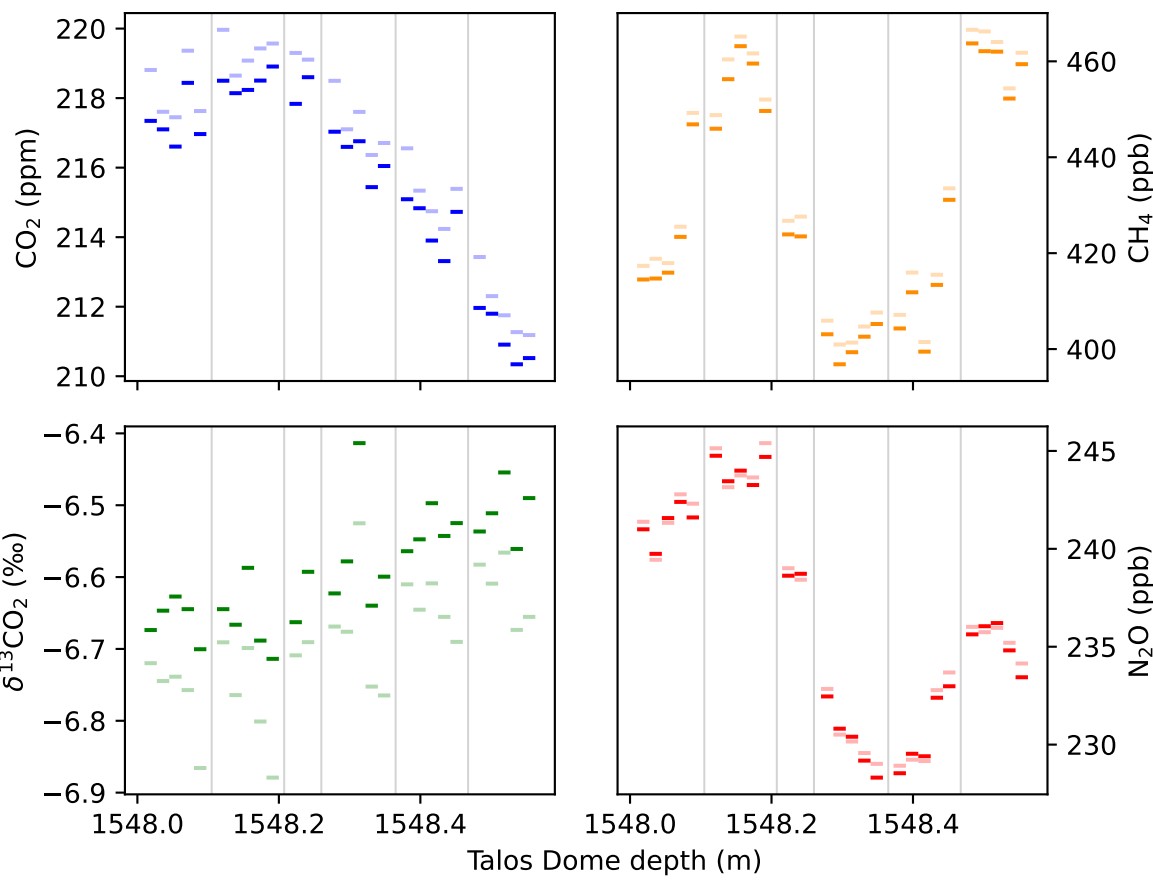

**Figure 11.** 57 cm of the Talos Dome ice core analyzed in ∼2 cm resolution with the described LISE/QCLS system. Faded colors show the raw data (but with calibration and drift correction), while solid colors denote the same measurements but with a dip tube specific correction for average offsets deduced from the standard over ice experiments. Vertical gray lines separate individual, ∼10 cm long ice samples, which were sublimated yielding typically five sub-samples. The shorter sample with only two sub-samples is due to a visible fracture at 1548.25 cm depth.

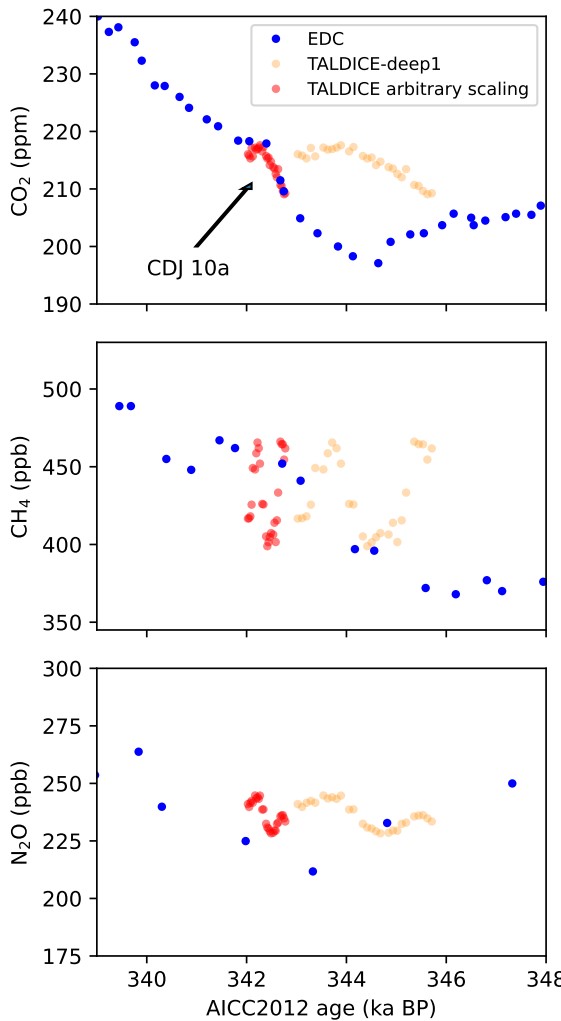

**Figure 12.** Comparing our high resolution Talos Dome data to EDC records. EDC $CO_2$ and $CH_4$ from Nehrbass-Ahles et al. (2020) including carbon dioxide jump 10a, and $N_2O$ (Spahni et al., 2005) are plotted on the AICC2012 age scale (Veres et al., 2013), which is also the basis for TALDICE-deep1 (Crotti et al., 2021). The Talos Dome data is shown both on the TALDICE-deep1 age scale as well as on an arbitrary scaling that optimizes the fit to the EDC record.

on the same sample. This recycling of sample gas will be tested in the near future; vii) The analysis is unaffected by drilling fluid contamination of the sample gas, and last but not least viii) the precision of $CO_2$ measurements but also of $\delta^{13}CO_2$ for such small samples sizes (15 g of ice) is significantly improved over previous approaches. While some of these qualities were achieved by previous measurement techniques, the combination of all of them is unique and makes the LISE/QCLAS approach especially suited for the analysis of deep, highly thinned ice as expected from the BE-OI drilling at Little Dome C.

*Author contributions.* LM, DB and FK designed and developed the LISE and the sample and standard introduction system for the QCLAS, under the guidance and with input from HF and JS. RW and CR provided mechanical and electronic support. LM, DB, and FK tested the system and performed the measurements. BB developed the QCLAS with input by BT, LE, JS and HF. HF managed and supervised the project. DB and LM prepared the manuscript with contributions from all authors.

*Competing interests.* The authors declare no competing interests.

*Acknowledgements.* This study is part of the ERC Advanced Grant "deepSLice" (667507) of H.F. This project has received funding from the European Research Council (ERC) under the European Union's Horizon 2020 research and innovation programme (grant agreement No 667507). Additional financial support has been provided by the Swiss National Science Foundation (SNF grant no. 200020B_200328). We gratefully acknowledge the reference measurements of the standard gases by Peter Nyfeler and Markus Leuenberger at the University of Bern.

**Appendix A: Additional photos**

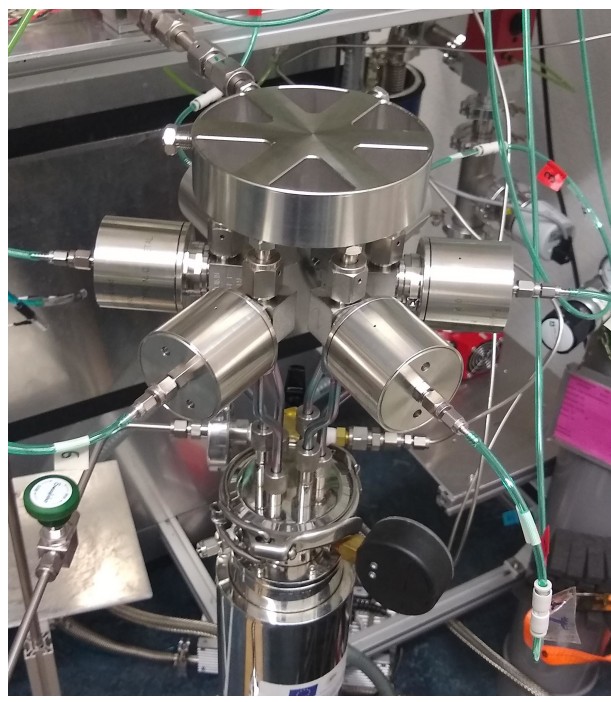

**Figure A1.** Custom aluminum vacuum manifold, termed Hexatubus, with six 14" dip tubes sitting in the cryostat, and connected to the extraction line at the top.

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
