# Peer review of "Laser-induced sublimation extraction for cm-resolution multi-species greenhouse gas analysis on ice cores"

_Atmospheric Measurement Techniques, 2022_

## Author Comment (AC1)

**Comments by Reviewer #1**

We would like to thank the Reviewers for their thoughtful and constructive comments that helped us to improve our manuscript. We addressed their comments individually and made revisions in response to their suggestions, as detailed below. The Reviewers' comments are highlighted in blue, followed by our response. Modifications to the manuscript text are reproduced in red.

The manuscript describes a new system for measuring major trace gases (CO2, CH4 and N2O) as well as d13C of CO2 from small sample amounts extracted through sublimation from ice core samples. The manuscript is well written and describes in detail the individual parts of the systems as well as the measurement procedures.

This system is a major step forward in the ability to analyse ice cores both in time resolution as well as precision.

Minor comments:

Abstract line 7: the vertical resolution depends on the cross section. Please add '(3.5x3.5cm cross section)'
Added.

Abstract line 16: Suggest to change to '...Pleistocene ice and highlighting the …'
Changed.

Page 2, line 27: Change to 'These limitations will..'
Changed.

Page2 line 31: Replace 'optimize' by 'increase'
Replaced.

Figure 1: This figure shows 'only' the inner parts of the system. It would be nice to have a sketch that includes all insulating, and cooling parts maybe in an appendix.
While we are happy to provide further technical details upon request, we do not see the benefit of a detailed sketch of all the plumbing in the manuscript. We believe that there is enough technical information in the manuscript for an experienced operator to replicate the system already without an explicit assembly plan.

Page 6, line 139: Is there a particular reason for choosing -100°C for the external trap?
The temperature of -100°C is high enough to not trap any of the studied greenhouse gases, but still low enough to let essentially no water vapour through. However, the value of -100°C is not a particular value, anything between -50°C and -120°C should work well.

Page 6, line 146: I am not clear how this manifold looks like. What part is made out of aluminium?
To support a better understanding of the setup, we added a photo of the manifold to the appendix.

Figure 3: Please add a legend explaining the symbols for pumps and gauges.
Added as suggested.

**Page 9, line 203: Milli-Q water is not gas free. It can under unfortunate conditions even be oversaturated in gases!**
We agree that Milli-Q water is not gas free and likely added a bit of gas to the measured air flow in Figure 4. However, even fully saturated water holds much less air than the bubble ice sections. Furthermore, most of that air would have been expelled during the freezing process. Given our deliberately minimal application of the Milli-Q water combined with its comparatively low air content we are confident that the measured air flow shown in Figure 4, and the vertical resolution derived from it, is only minimally affected by the Milli-Q glue. Having said that, we would not recommend the same approach if we were to actually analyze the composition of the sublimated sample gas.

We added: *Even though Milli-Q water is not gas free, its contribution to the measured air flow is minimal.*

**Page 9, lines 210 and following: I observe (Figure 4) that 1) The flow does not drop to 0 in neither the 1cm nor the 2 cm experiment. In the 1cm experiment the minimum level reached is about 0.1microliter/s and about half of that in the 2 cm experiment. There is no obvious trend over the course of the experiment. I agree with the conclusion that the side sublimation is limited to some centimetres below the sublimation front. However, it seems to me that this reaches way beyond 2cm. The contribution to the flux is still a couple of % which may cause a problem when different samples are measured.**
If the side sublimation were to reach way beyond 2 cm, we should see a decreasing trend of the minimum level (baseline) observed throughout the sublimation, especially for the last minimum in the 1 cm experiment, where there is no more bubbly ice underneath that could contribute to the measured air flow by side sublimation. But the observed minima are stable throughout both runs, which is consistent with the assumption that side sublimation does not contribute significantly beyond 2 cm. We do acknowledge the Reviewer's point that a small contribution of side sublimation from the lower layers is likely still happening and may cause some smearing of the measured data. We added the following clarifying text to the manuscript:

*While we have shown that the contribution of side sublimation from lower layers is small (a few % at most), it may still lead to some smoothing across the sub-samples if the concentration gradient in a sample is large.*

**Page 12, line 264: Replace 'produced' with 'prepared'**
Replaced, also in another instance where produced was used in combination with standard gases.

**Page 12, line 267 and following: Is there any evidence that the consecutive cleaning of the extraction vessel makes a difference.**
It is rather difficult to determine exactly which part of the procedure is responsible for how much of the outcome without a lot of testing. We obtained good and reproduceable results with the consecutive cleaning, but we don't have enough experiments with single cleaning to conclude that the 2$^{nd}$ cleaning step is crucial in the process. Thus, we can only describe what worked for us.

**Page13, line 3: I find it hard to believe that the Ar completely hinders laboratory air from entering. I suggest replacing 'hinders' to 'limits' or 'reduces'**
Agreed. We adopted the suggested wording.

**Page 13, line 296: Is there any evidence that 10 min is long enough to reach equilibration with the surfaces?**
Again, it is difficult to say with our measurements and experiments. We did some tests at the

beginning with anywhere from 3 to 15 min and it did not seem to make a difference at all. But it could be that equilibration with the surfaces is small enough that even if one does not reach full equilibration it wouldn't show up in the results. We are confident in saying that wasting the sample gas into the turbo pump for longer than 10 mins wouldn't improve the described biases of the system.

We added a sentence: *A longer pumping time did not improve the results and leads to unnecessary loss of sample.*

Figure 6: Please label Std#6
Done.

Section 3.2: You write 'This standard is used later to pointwise correct any offset in the concentration values.' Later you write of 'bracketing values'. That implies that you correct by using standard measurements from before and after the sample but it does not look like you do that. Therefore, I suggest you use a different term than 'bracketing' to describe your procedure.
We have changed the wording from 'bracketing' to 'pointwise drift correction' throughout the manuscript.

Page 20, lines 410 and following on sample 768.35: This sample shows higher variability than the others except for CH4. Clathrate formation/decay should not have an effect, since you extract 100% of the sample. Gas loss should affect all gases also CH4. I am unclear what your explanation for the higher variability in the specific sample is and why it is not better than from previous systems.
We missed to mention here that in addition to fractionation and diffusive transport between coexisting bubbles and clathrates (on the scale of μm) the Lüthi *et al*. paper discusses diffusion of gases over longer distances (mm to cm) that leads to enhanced variability of $CO_2$ concentrations in the bubble-to-clathrate-transition zone (BCTZ), a sort of layering. In this case, higher resolution measurements are expected to yield higher variability as it is a real feature in the ice and even 100 % extraction efficiency won't eliminate that. In the long run (below the BCTZ), diffusion eventually smoothens the cm variability again. We changed the wording explaining the BCTZ sample to clarify this point:

*The sample at 768.35 m depth is an outlier in this regard with a much higher intra-sample variability, likely due to the fact that this sample lies in the bubble-to-clathrate transition zone (Neff, 2014), where layered early clathratization occurs and differently fast permeation rates between bubbly layers and clathrate layers for different gas species lead to the cm scale variability in the gas composition (Lüthi et al., 2010; Oyabu et al., 2022).*

---

## Author Comment (AC2)

**Comments by Reviewer #2**

We would like to thank the Reviewers for their thoughtful and constructive comments that helped us to improve our manuscript. We addressed their comments individually and made revisions in response to their suggestions, as detailed below. The Reviewers' comments are highlighted in blue, followed by our response. Modifications to the manuscript text are reproduced in red.

Machler et al describe a new analytical technique and system for analysis of trace gases in ice cores via laser induced sublimation combined with quantum cascade laser absorption spectroscopy (QCLAS). In my opinion, this system represents a very large advance in ice core trace gas analysis. The analytical precisions are on par with some of the best prior techniques, but the prior techniques required similar or (sometimes much) larger sample sizes for analysis of a single species, whereas here CO2, CH4, N2O and d13CO2 are analyzed simultaneously. The capability of the system in terms of depth resolution is very impressive and is better than what continuous flow analysis (CFA) systems can currently do for trace gases. The system seems very well suited and ready to be used for the stated purpose of generating high-resolution records from the Beyond EPICA – Oldest Ice Core. There is an impressive range of technical innovations / excellent design ideas that have been incorporated into the system. The testing of system is very thorough. The paper is very well written and the results are clearly presented. My only real suggestion for further improvement is to conduct a series of tests with Holocene ice from a relatively high-accumulation ice core where the trace gas species are very well characterized and there is high confidence that no significant changes in concentrations / isotopes are expected. Overall, the authors are to be congratulated on a fantastic new system / technique / manuscript. Below are some very minor suggestions / typo corrections:

The suggested additional measurements on Holocene ice would be certainly possible, obviously with considerable time and effort, but we don't think that this is absolutely essential for the method assessment. If the aim is to further test reproducibility then a low accumulation core such as EDC should be better suited as it has a wider gas age distribution and thus less expected variability on small scales. On the other hand, if the additional measurements are to serve a better comparison to existing records (to judge accuracy at the ppm level), we run into well-known issues between different extraction techniques (e.g. for $N_2O$ concentration) and yet unresolved offsets between $CO_2$ records from different cores, all of which frustrate a meaningful comparison to other records. Overall, given the time constraints and the (in our opinion) lack of absolute necessity for the suggested additional measurements, we believe that the presented data should suffice to confirm the measurement performance in its current form. We also point out that first publications to reconstruct real ice core records using this technique will follow this methodological paper, which will give ample opportunity to investigate reproducibility and replication of previous records. However, if the editor deems it important enough to delay publication of this paper for it, we are happy to do these measurements as soon as practical. As of the three people who have run the system in the past two have left the lab and the third is in Antarctica for fieldwork, the measurements could be performed in April/May at the earliest.

Line 39 replace "shortly" with "briefly"
Replaced.

Line 108 change to "sealed glass vessel with a flat base and lid"
Changed.

Figure 3: include a legend explaining all symbols
Added as suggested.

Line 161 "combi" ï☐ "combination"
Corrected.

Line 201 "the Byrd ice core"
Changed.

Line 220 change to "…increase or decrease…"
Changed.

Figure 5. Clarify: was the photo on the left from a sublimation with P < 0.15 hPa?
We changed the figure caption to: *Ice after 4 cm of sublimation. Penitentes in the right picture occur at pressures of more than 0.15 – 0.2 hPa measured at the top of the extraction vessel, while on the left the pressure was kept below 0.15 hPa.*

Line 270 "from the inner surface"
Changed.